# Sub-Regional Variation and Characteristics of Cabernet Sauvignon Wines in the Eastern Foothills of the Helan Mountain: A Perspective from Phenolics, Visual Properties and Mouthfeel

**DOI:** 10.3390/foods12051081

**Published:** 2023-03-03

**Authors:** Bing-Yan Zhao, Xin-Ke Zhang, Yi-Bin Lan, Chang-Qing Duan, Bao-Qing Zhu, De-Mei Li

**Affiliations:** 1Department of Food Science and Engineering, College of Food Science and Engineering, Beijing University of Agriculture, Beijing 102206, China; 2“The Belt and Road” International Institute of Grape and Wine Industry Innovation, Beijing University of Agriculture, Beijing 102206, China; 3Center for Viticulture and Enology, College of Food Science and Nutritional Engineering, China Agricultural University, Beijing 100083, China; 4Key Laboratory of Viticulture and Enology, Ministry of Agriculture and Rural Affairs, Beijing 100083, China; 5Beijing Key Laboratory of Food Processing and Safety in Forestry, Department of Food Science, College of Biological Sciences and Biotechnology, Beijing Forestry University, Beijing 100083, China

**Keywords:** sub-region, Cabernet Sauvignon, phenolic compounds, CATA, QDA

## Abstract

As one of the most promising wine regions in China, the eastern foothills of the Helan Mountain (EFHM) in the Ningxia Hui Autonomous Region has attracted great attention recently. Geographically, EFHM is divided into six sub-regions, namely Shizuishan, Xixia, Helan, Qingtongxia, Yongning and Hongsipu. However, there have been few reports on the character and differences between wines in the six sub-regions. In this experiment, a total of 71 commercial Cabernet Sauvignon wines from six sub-regions were collected, and their phenolic compounds, visual properties and mouthfeel were investigated. The results showed that wines from the six sub-regions of EFHM showed distinctive phenolic profiles and could be distinguished through the OPLS-DA mode using 32 potential markers. In terms of color, Shizuishan wines showed higher *a** values and lower *b** values. The sensory evaluation showed that Hongsipu wines had higher astringency strength and lower tannin texture. The overall results implied that the phenolic compounds of wines in different sub-regions were affected by terroir conditions. To the best of our knowledge, this is the first time that a wide coverage of phenolic compounds has been analysed for wines from the sub-regions of EFHM, which could provide valuable information in deciphering the terroir of EFHM.

## 1. Introduction

Wine is the most popular alcoholic drink across the world because of its unique culture and complex flavor. The term ‘terroir’ appears to have first been applied in the 14th century in France and encompasses the key natural elements of landscape features, soil characteristics, climate and human factors (socio-economics, history, genotype, variety and rootstock selection, winemaking technologies and vineyard management practices) that result in the production of unique, site-specific terroir wines [1,2].

The eastern foothills of the Helan Mountain (EFHM) in the Ningxia Hui Autonomous Region, with latitude of 34°14′–39°23′ and longitude of 104°17′–107°39′, is located in the flood plain zone between the alluvial plain of the Yellow River and the alluvial fan of Helan Mountain (Figure 1). EFHM has a classic continental climate with sufficient sunlight, heat and suitable rainfall (2850–3110 h of sunshine, 3100–3500 °C of active accumulated temperature and 150–200 mm of rainfall). Helan Mountain obstructs cold air from the northwest, and the irrigation canals of the Yellow River in the east provide sufficient water, making EFHM a suitable region for wine grapes [3,4]. In 2011, the General Administration of Quality Supervision, Inspection and Quarantine of China ratified the protection of place of origin (POD) of EFHM in Ningxia Hui Autonomous Region, making it the third Appellation wine region, after Changli in Hebei province and Yantai in Shandong province.

High in the north and low in the south, the Helan Mountain borders the desert in the south, creating a microclimate that varies from north to south. In addition, the wine-producing region in EFHM is located along the mountain and river, and this provides different soil types due to various distances from the mountain, thus creating a variety of vineyard conditions. After 10 years of development, delicate sub-regions in EFHM have been formed and recognised officially, namely Hongsipu region, Qingtongxia region, Yongning region, Xixia region, Helan region and Shizuishan region (Figure 1). Although EFHM is the first appellation in China with geographical indication for sub-regions, the comprehensive understanding of the terroir in these sub-regions is far from adequate due to a short development period.

Cabernet Sauvignon is the offspring of a cross between Cabernet Franc and Sauvignon Blanc [5], originated from the Bordeaux region in France, and is regarded as one of the most important grape varieties for making high quality red wines [6]. It has pronounced influences on the wine making regime, given its aptitude to vinification by itself, but particularly when blended with other grape varieties [7]. It is also one of the most popular grape varieties in China and has a sizable production in EFHM. EFHM contains 4000 hectares of Cabernet Sauvignon vine, accounting for 63% of the total wine grape cultivation area in the region. In addition, Cabernet Sauvignon is grown in all sub-regions, accounting for 50% or more of the total wine grape area in each. Cabernet Sauvignon is a late maturing variety with a very thick pericarp, resulting in abundant accumulation of phenolic compounds. Color and properties are important organoleptic aspects of wines, and phenolic compounds contribute to color [8,9], bitterness and astringency [10,11]. The phenolic composition of a wine is dependent on cultivar [12,13], climate [14], soil type [15], viticulture practice and vinification techniques [16], all of which are factors of terroir.

Different wine regions can have different terroir conditions which can lead to great differences in the quality, style and composition of wine. This difference could be reflected in the profile of secondary metabolites especially aromatic and phenolic compounds, as well as sensory characteristics. For instance, regional variation was characterized in 14 commercial Canadian Riesling wines using descriptive analysis (DA), and significant differences in the key aroma compounds of the wines were found [17]. Similarly, regional variation was notably observed based on total polyphenols, trans and cis-resveratrol and biogenic amines in 73 wines from four Southern Italy regions [18]. Using 18 non-flavonoid phenolic compounds, the origin of 43 Riesling wines from five regions in the Czech Republic could be successfully distinguished [19]. Li et al. analysed the phenolic compounds in Cabernet Sauvignon wines from five distinct regions across China and found different phenolic profile in wines from hot and arid regions in northwest China and warm and humid regions in eastern China, respectively. Compared with wines from the other four regions selected for the experiment, wines from Deqin in Yunnan province (a highland valley in southwest China) contained extremely high concentrations of cyanidin derivatives and quercetin derivatives, but extremely low concentration of epicatechin, reflecting the terroir effect of the wines [20].

However, the characteristics of phenolic compounds in Cabernet Sauvignon wines from different sub-regions in EFHM have not been reported.

With the aim of studying the delicate regional variation of Cabernet Sauvignon wines, 71 wines from six sub-regions of EFHM were selected. Primary phenolic compounds, including non-anthocyanin phenolics, anthocyanins and their derivatives, were analysed by high-performance liquid chromatography-triple-quadrupole (HPLC-QqQ-MS/MS). The colors of the wines were quantified using the CIELAB method. At the same time, the mouthfeel of the wines was evaluated using Check-All-That-Apply (CATA) and Quantitative Descriptive Analysis (QDA). To the best of our knowledge, this is the first time that such a wide range of wines from six sub-regions of EFHM has been collected. The phenolic profile of these samples from such a wide coverage could provide valuable information in deciphering the terroir of EFHM, which in turn could provide an academic basis for the better development of the wine industry in China.

## 2. Materials and Methods

### 2.1. Wine Samples

A total of 71 commercial wines from six different sub-regions of EFHM were collected. These wines were all Cabernet Sauvignon or Cabernet Sauvignon-dominant blends (>75%), with vintages ranging from 2015 to 2021. The basic wine compositions (residual sugar, alcohol level, pH, volatile acidity and total acidity) were measured using a WineScan (FT 120) rapid-scanning infrared Fourier-transform spectrometer with FOSS WineScan software version 2.2.1 (Foss Electric, Hillerød, Denmark). Information for all wine samples is given in Appendix A.

### 2.2. Chemicals and Standards

Methanol, formic acid and acetonitrile (HPLC grade) were purchased from Shanghai Macklin Biochemical Co., Ltd. (Shanghai, China). Deionised water was purchased from Wahaha Co., Ltd. (Hangzhou, China). The standard compounds of anthocyanin and non-anthocyanin phenolics were purchased from Sigma-Aldrich (St. Louis, MO, USA), ChromaDex (Irvine, CA, USA), and Extrasynthese (Genay, France).

### 2.3. Analysis of Phenolic Compounds

Phenolic compounds in wines were analysed according to various published methods. All wines were filtered through a 0.22 μm inorganic polyether-sulfone membrane prior to HPLC-MS analysis. An Agilent 1200 series high-performance liquid chromatographer equipped with an Agilent 6410B triple-quadrupole (QqQ) mass spectrometer (Agilent Technologies, Santa Clara, CA, USA) was used. The column was a Poroshell 120 EC-C18 column (150 mm × 2.1 mm, 2.7 µm; Agilent Technologies, Santa Clara, CA, USA). The mobile phases A were 0.1% formic acid in water; the mobile phases B were 0.1% formic acid in methanol and acetonitrile (50/50 *v/v*).

#### 2.3.1. Analysis of Non-Anthocyanin Phenolic Compounds [21]

The gradient elution was: (1) from 10% to 46% B in 28 min; (2) from 46% to 10% B in 1 min. The post time was 5 min. The injection volume was 5 µL and the flow rate was 0.4 mL/min. The column was thermostatically controlled at 55 °C. An electrospray ionization source was used with 4 kV voltage and in the negative mode. The temperatures of the ion source and the drying gas (N2) were 150 °C and 350 °C, respectively. The drying gas flow rate was 12 L/min and the nebulizer pressure was 35 psi. The precursor ions and product ions of each phenolic compound were set in the multiple reaction monitoring (MRM) mode according to the published method [22]. The quantification of each phenolic compound was achieved through a calibration curve for each commercially available phenolic standard.

#### 2.3.2. Analysis of Anthocyanins [23]

The HPLC conditions for the analysis of anthocyanins were the same as for non-anthocyanin phenolic compounds. The ion source parameters were the same as for the non-anthocyanin phenolics, except that the positive mode was used. The MRM mode was also selected for the detection of anthocyanins according to a previous publication [23]. The quantification of each anthocyanin compound was achieved through the malvidin-3-*O*-glucoside calibration curve. 

#### 2.3.3. Analysis of Anthocyanin Derivatives [24]

The mobile phase for the analysis of anthocyanin derivatives was the same as the above. The gradient elution started with the isocratic elution of 100% A for 1 min, then linearly increased B to 25% at 3 min, to 30% B at 15 min, to 100% B at 20 min, when the column was maintained by eluting with 100% B for an additional 5 min. The post time was 5 min. The injection volume was 10 µL and the flow rate was 0.3 mL/min. The ion source conditions were the same as for the analysis of anthocyanins. The MRM parameters for the anthocyanin derivatives were set according to a previous publication [24]. The semi-quantification of each anthocyanin derivative was calculated from the basis of the calibration curve of malvidin-3-*O*-glucoside measured by the same method.

### 2.4. Color Measurement

The chromatic characteristics of all wines were quantified using the CIELAB approach [25]. All wines were first filtered through a 0.22 μm inorganic polyether-sulfone membrane and then placed in a 2-mm-optical-path glass cuvette. The absorbances at wavelengths ranging from 400 nm to 700 nm (at 1 nm intervals) were measured using a UV-visible spectrophotometer (Shimadzu UV-2450, Shimadzu Co., Kyoto, Japan). The values of lightness (*L**), red-greenness (*a**), and yellow-blueness (*b**) were calculated accordingly.

### 2.5. Sensory Analysis

It should be emphasized that this study complied with The Code of Ethics of the World Medical Association (Declaration of Helsinki), and all sensory evaluators provided informed consent to participate in the study. The Research Ethics Committee of China Agricultural University gave its approval for human subjects to be involved in this study, reference number CAUHR-20220901.

#### 2.5.1. CATA

CATA is a rapid descriptive analysis method for consumers to select all sensory properties from a given list of sensory descriptors. CATA chooses consumers to replace professional sensory evaluators, without the need for professional training and maintenance [26]. Prior to the formal experiment, all 71 wine samples were evaluated by 16 experienced experts (professional sommeliers, winemakers and faculty), including 10 males and 6 females, aged between 26 and 58 years. Firstly, twelve experts were asked to participate in two sessions, each of which was divided into two rounds (approximately 50 min each). A glossary was then created. In the subsequent session, the appropriate terms were then agreed on by another four experts who had evaluated 71 wines. The final list consists of the 18 descriptors in Appendix A. 

Forty people were randomly recruited to participate in the CATA, 14 males and 26 females, aged between 20 and 30 years old. All sensory evaluators had experience of wine tasting and were selected on the basis of their interest. Prior to the formal CATA, each evaluator was trained to successfully describe the astringency and tannin texture of wines. For the training session, gradient solutions of skin tannin extract (0.1, 0.5, 1.0, 1.5 and 2.0 g/L) were used, and the scales for perceived astringency strength were set as weak, moderately weak, moderate, moderately strong and strong, respectively. For the descriptors describing the tannin texture sensations (satin, velvet, fine emery and abrasive), the touch of a physical standard with the fingertips could be used as a reference, as recommended by Gawel et al. [27], e.g., the touch of a velvet cloth to represent the mouth surface sensation labelled velvet. The detailed sensory options are shown in Appendix A. After the training, 40 sensory evaluators were instructed to taste prepared wine samples and then check the appearance, astringency strength and tannin texture options using a pre-designed questionnaire. Water and tasteless biscuits were prepared for each evaluator, and they were requested to relieve their mouths after each tasting. The entire CATA was conducted in 3 sessions, and each session consisted of 4 rounds. In each round, 6 wines were served and the evaluator was requested to complete the questionnaire within 25 min. A 10-min break was provided after the first two rounds. All wines were prepared in International Standards Organisation (ISO) wine tasting glasses (ISO 3591:1977) containing approximately 30 mL of wine and presented in a random order. All sensory evaluators worked in individual booths at a controlled temperature (20 °C).

#### 2.5.2. QDA

QDA is one of the classical descriptive sensory techniques to describe the characteristic and the intensity of sensory properties from a single evaluation of a product [28,29]. It has been widely applied to vegetables [30], milk [31], wine [32], etc. CATA can only identify the characteristics of wines from different sub-regions, but cannot quantify these characteristics, especially when comparing the differences between wines from different sub-regions. Therefore, based on the results of CATA, wines with typical characteristics of each sub-region were selected for QDA. Seventeen experienced sensory evaluators were invited to participate in the QDA, 12 females and 5 males, aged between 25 and 32.

Each sensory evaluator had passed the selection, training and periodic testing stipulated in the national standards of China (GB/T 16291.1-2012). Prior to the formal experiment, two Cabernet Sauvignon wines were used to standardize the scoring criteria and all sensory evaluators were requested to evaluate and discuss the body, finish, astringency strength and tannin texture of the wines until they reached a consensus. They were then asked to rate the selected wines on a scale of 0 to 10 for the four mouthfeel characteristics, with 0 being very weak and 10 being very strong. The wines for the QDA were divided into four rounds for scoring, lasting a total of two hours, with a 10-min break at the end of each session. The environment and supplies of QDA are the same as for CATA conditions.

### 2.6. Statistical Analysis

The identification and quantification of all phenolic compounds in the wines were achieved using Mass Hunter workstation software (version 10.0) (Agilent Technologies, Santa Clara, CA, USA). One-way analysis of variance (ANOVA) was conducted and Duncan’s post-hoc test with a significance level of 0.05 was performed in SPSS Statistics software (version 25.0) (IBM, Chicago, IL, USA). Soft Independent Modeling of Class Analogy (SIMCA, version 14.1 from Umetrics) was used for Orthogonal Partial Least Squares Discriminant Analysis (OPLS-DA). Hierarchical cluster analysis was achieved through “MetaboAnalyst 5.0” (http://www.metaboanalyst.ca/, (accessed on 14 October 2022)). CATA data were analysed using XLSTAT 2019 (Addinsoft, New York, NY, USA).

## 3. Results and Discussion

### 3.1. Basic Wine Compositions

The basic compositions of all the wines, including ethanol level, residual sugar, pH, total acidity and volatile acidity are listed in Appendix A. The alcohol level of all the wines ranged from 13.14% to 15.74% (*v*/*v*), the residual sugar ranged from 1.7 to 10.3 (g/L), the pH ranged from 3.54 to 4.23, the total acidity ranged from 4.9 to 7.6 (g/L, tartaric acid equivalent), and the volatile acidity ranged from 0.5 to 1.0 (g/L). All the basic parameters of the wines conformed to the national standards of China (GB/T 15037-2006) and can be used for subsequent analysis.

### 3.2. OPLS-DA Analysis

A total of 67 phenolic compounds were identified in all wines by HPLC-QqQ-MS/MS. Detailed information is shown in Table 1. The OPLS-DA model was used to differentiate and characterize six sub-regions in EFHM, as it has shown good performance in wines with subtle regional and vintage variations [33]. In the model, R2 is a measure of fitness, i.e., how well the model fits the data. Q2 indicates the predictability of the model. As shown in Figure 2A, a separation was obtained by a reliable OPLS-DA model (R2X = 0.662, R2Y = 0.331, Q2 = 0.159) based on the concentration of phenolic compounds. The further validation of the model was tested by 7-fold internal cross-validation and 200-time permutation tests. The prediction result of the cross-validated score plot was basically consistent with the actual score plot (Appendix A and Figure 2A), indicating a good predictive effect. The results of 200-time permutation tests showed that the OPLS-DA model did not overfit (Appendix A). According to the model, it can be clearly seen that the wines from Yongning, Qingtongxia and Hongsipu region were all well separated. Meanwhile, there was some overlap between the wines from Helan and Xixia region (Figure 2A). We speculated that this may be due to the fact that Helan and Xixia regions are adjacent to each other (region 2 and 3 in Figure 1), leading to more similarities in topographical features. Interestingly, wines from Shizuishan and Qingtongxia regions also overlap in the model, but the two sub-regions have nothing in common other than the same altitude (Appendix A).

Figure 2B shows the correlation between the explanatory variables, i.e., the concentration of phenolic compounds (in purple dots) and the dependent variables, i.e., the sub-regions (in black dots) in the first and second principal components. These compounds, located close to the sub-regions in Figure 2B, could be considered as potential features with these sub-regions. Figure 2C shows the Variable Importance for the Projection (VIP) plot of the OPLS-DA model. A higher VIP value indicates a greater contribution of the explanatory variable to the discriminative ability in OPLS-DA. Normally, a VIP value above 1 could be considered as a threshold for selecting potential markers [34]. A total of 32 phenolic compounds with VIP above 1 were screened (Figure 2C). 

Combined with Figure 2B,C, it can be seen that cyanidin-3-*O*-acetylglucoside (Cy-Aglu), malvidin-3-*O*-glucoside-(epi)catechin (A type) (Mv-(e)cat) and (epi)catechin-malvidin-3-*O*-glucoside (B type) ((E)cat-Mv) were the potential markers in wines from Xixia region. Delphinidin-3-*O*-glucoside-pyruvic acid (Dp-py), petunidin-3-*O*-glucoside-pyruvic acid (Pt-py) and cyanidin-3-*O*-glucoside (Cy-glu) were the potential markers in wines from Helan region and Hongsipu region. Delphinidin-3-*O*-glucoside-acetaldehyde (Dp-ace), cyanidin-3-*O*-glucoside-acetaldehyde (Cy-ace) and protocatechuic acid (PA) were the potential markers in wines from Yongning region.

### 3.3. Sub-Regional Variation of Phenolic Compounds

#### 3.3.1. Comparison of Non-Anthocyanin Phenolic Compounds

A total of 28 non-anthocyanin phenolic compounds were identified in all wines via HPLC-QqQ-MS/MS (Appendix A), including six flavan-3-ols, nine phenolic acids (three hydroxycinnamic acids and six hydroxybenzoic acids) and 13 flavonols. 

It was found that the concentration of the total detectable flavan-3-ols in wines from six sub-regions ranged from 145.12 to 673.67 mg/L. Wines from the Yongning region had the lowest flavan-3-ol concentration, while those from the Hongsipu region had the highest. (Appendix A). Previous studies had shown that the type of soil has a significant effect on phenolic compounds in grape [35,36]. The soil of Hongsipu region is a sierozem soil with loose texture and good aeration, which makes the soil have strong water and fertilizer holding capacity and rich calcium concentration [37,38]. Such soil provided favourable conditions for tannin accumulation [39]. This may be one of the reasons accounting for the higher concentration of flavan-3-ols in wines from Hongsipu region.

The total detectable flavonol concentrations in wines from the six sub-regions ranged from 21.74 to 118.69 mg/L. The concentrations of myricetin-glucoside (M-glu), quercetin-glucoside (Q-glu) and isorhamnetin-glucoside (I-glu) in Helan wines were significantly higher than those in other sub-regions, resulting overall in the highest total flavonol concentrations. In contrast, the Hongsipu wines had the lowest flavonol concentrations (Appendix A). Both genotype (variety) and environment are critical factors in controlling the production of flavonols [40]. Furthermore, in the case of wines, even common wine-making processes, including grape skin contact, stabilization processes and ageing, have been shown to cause significant changes in flavonols [41]. However, the reason for the high concentration of flavonols in Helan wines is still unclear and further studies are required.

The concentration of total detectable hydroxybenzoic acids ranged from 10.16 to 51.38 mg/L, with the highest concentration in Shizuishan wines and the lowest in Qingtongxia wines. Gallic acid (GLA) was the predominant hydroxybenzoic acid (6.77 to 38.12 mg/L), which was consistent with previous reports [42]. Compared with the wines from other sub-regions, the wines from the Hongsipu region contained a significantly higher concentration of GLA but a lower concentration of vanillic acid (VA), 4-hydroxybenzoic acid (4-HBA) and PA. In contrast, more 4-HBA and PA were detected in Shizuishan wines, even up to about twice as much as in wines from other sub-regions (Appendix A).

In terms of hydroxycinnamic acids, concentration ranged from 4.47 to 29.19 mg/L. In addition, as the predominant hydroxycinnamic acid [43,44], caffeic acid (CFA) showed no significant difference among the six sub-regions. More generally, the remaining hydroxycinnamic acids in the six sub-regions of wines were not significantly different except for 3-hydroxycinnamic acid (3-HCA) (Appendix A).

#### 3.3.2. Comparison of Anthocyanins

A total of 15 anthocyanins were identified by HPLC-QqQ-MS/MS in all wines, including non-acylated anthocyanins, acylated anthocyanins and coumaroylated anthocyanins (Appendix A). In general, there were no significant differences in the total concentration of detectable anthocyanins between the six sub-regions. Nonetheless, wines from Helan and Shizuishan region showed a higher and lower concentration of total detectable anthocyanins, respectively. The biosynthesis of anthocyanins in grape was influenced by temperature, with a lower temperature promoting the expression levels of anthocyanin biosynthesis genes such as VIMYBA2, while a higher temperature may suppress them [45,46]. Although climate data for recent years were not available, a higher effective accumulated temperature from July to October was observed in the Shizuishan region from historical data (Appendix A). This may be one of the reasons that account for a lower anthocyanin concentration in this region.

All five types of anthocyanins were detected: cyanidin, delphinidin, peonidin, petunidin and malvidin. The proportion of the five types of anthocyanins in wines from the six sub-regions was almost the same (Appendix A). Cyanidin-type anthocyanins had the lowest concentration and malvidin-type anthocyanins had the highest concentration, suggesting that the anthocyanin composition was not influenced by sub-regional factor but might be inherently controlled by a genetic factor, such as cultivar [47]. Among all the anthocyanins, malvidin-3-*O*-glucoside (Mv-glu) showed the highest concentration in wine, in agreement with previous results [48,49].

#### 3.3.3. Comparison of Anthocyanin Derivatives

A total of 24 anthocyanin derivatives were identified in all wines by HPLC-QqQ-MS/MS (Appendix A), including three direct flavanol-anthocyanin condensation products (F-A), five direct anthocyanin-flavanol condensation products (A-F) and 16 pyrano-anthocyanins (10 vitisins, three flavanyl-pyrano-anthocyanins and three pinotins). Overall, there was no significant difference in the concentration of total detectable anthocyanin derivatives in different sub-regions, but it could be seen that the wines from Hongsipu region had the highest concentration of total detectable anthocyanin derivatives, followed by the wines from Helan region, and the wines from Shizuishan region had the lowest concentration of total detectable anthocyanin derivatives (Appendix A). For most anthocyanin derivatives, such as delphinidin-3-*O*-glucoside-pyruvic acid (Dp-py) and petunidin-3-*O*-glucoside-pyruvic acid (Pt-py), their concentrations were significantly higher in wines from Hongsipu, which was largely consistent with the concentration conditions of their anthocyanin precursors. For those anthocyanin derivatives with no significant difference, a reasonable explanation was that there were also no significant differences in the concentrations of their anthocyanin precursors. In addition, anthocyanin derivatives were mostly formed during the process of alcoholic fermentation, malolactic fermentation and aging of wine [50]. For example, the precursors of vitisins were pyruvic acid and acetaldehyde derived from yeast metabolism during alcoholic fermentation [51]. Therefore, the accumulation of anthocyanin derivatives in wines from the six sub-regions was a complex process influenced by many factors, such as wine-making technology and grape variety.

### 3.4. Hierarchical Cluster Analysis

The phenolic compounds of the 71 EFHM wines were analyzed using hierarchical cluster analysis to determine the similarities for these sub-regions (Figure 3). Yongning region, Helan region, Qingtongxia region, Shizuishan region and Xixia region were consecutively grouped into one category, which may be due to the fact that these sub-regions are all adjacent to the foothill of Helan Mountain and therefore share similar climate and soil type. As the southernmost sub-region of EFHM, Hongsipu region is far away from other sub-regions and less protected by Helan Mountain (Figure 1). Moreover, its high altitude makes it more susceptible to the northwesterly cold flow. It also received more rainfall than most of the sub-regions (Appendix A). These factors led to large differences in terroir, so Hongsipu region was divided into a separate category, which is consistent with the clustering results of different sub-regions in EFHM by Zhang et al. [52].

### 3.5. Comparison of Color of Wines

The chromatic properties of wines from different sub-regions were quantified using the CIELAB approach (Table 2). The color of wine is originally derived from anthocyanins extracted from grape skins during winemaking. Anthocyanins showed a negative correlation with *L** and *b** values but a positive correlation with *a** values [53]. The results of the chromatic characteristics of all wines showed no significant differences in *L** values between wines from different sub-regions, indicating no significant variation in color intensity. Wines from Shizuishan had significantly higher *b** values (more yellowness) and lower *a** values (less redness) than those from other sub-regions. Based on the quantification of phenolic compounds, we found that this might be due to the lowest concentration of total anthocyanins in Shizuishan wines (Appendix A). In addition, wines from Shizuishan region had a significantly higher pH than those from other sub-regions (Appendix A), while higher pH could result in wine losing its color intensity and redness [54,55].

### 3.6. Sensory Characteristics of Wines

CATA was used to characterize the appearance, astringency strength and tannin texture of Cabernet Sauvignon wines from the six sub-regions of EFHM. Correspondence analysis (CA) is a multivariate statistical technique which is applicable to tables of categorical data [56]. In the study, CA was used to explore the visual and mouthfeel characteristics of wines in descriptors of specific characteristics such as appearance, astringency strength and tannin texture. In CA, there was a significant difference in frequency for 17 of the 18 descriptors (*p* < 0.05) and their correlation with the wines is shown in Figure 4A. F1 and F2 explained a 56.59% of the total variance. Wines in the first quadrant were deep ruby or deep purple in appearance, with strong to moderately strong astringency strength and fine emery or abrasive tannin texture. Wines in the second quadrant were brick red, brown or garnet in appearance. In the third quadrant, the wines were mainly light ruby or ruby in appearance, with moderately weak or weak astringency strength and satin or velvet tannin texture. In the fourth quadrant, the wines were mainly light purple and purple in appearance.

In addition, Appendix A shows the frequencies of the descriptors used to describe the wines, from which frequencies above 20% were selected as representative sensory characteristics. The typical visual and mouthfeel characteristics of wines from different sub-regions were obtained. The results showed that the wines from Shizuishan region were ruby, while the astringency strength was moderately weak, with velvet or fine emery tannin texture. However, moderately strong astringency strength was also selected by some sensory evaluators for Shizuishan wines. Most Helan wines were deep ruby or ruby, and their astringency strength was considered as moderately weak, with velvet or fine emery tannin texture. Wines from Xixia region were ruby or deep ruby in appearance, with moderately weak astringency strength and velvet or fine emery tannin texture. Yongning wines were light ruby or deep ruby, some wines were described as garnet, and the astringency strength was moderately weak, with velvet or fine emery tannin texture. Qingtongxia wines were mostly ruby, and the astringency strength was moderately weak or weak, felt satin or velvet, and a few were described as fine emery. Hongsipu wines were purple or ruby in appearance, with moderately weak astringency strength and fine emery tannin texture.

Using the above-mentioned criteria (frequency > 0.2), 27 representative wines from different sub-regions (excluding Shizuishan region) were used for QDA. The box plot in Figure 4B shows the profile of astringency strength, tannin texture, body and finish of wines from the different sub-regions. The results shows that the astringency strength of Hongsipu wines is significantly higher (Appendix A). It has been confirmed that flavan-3-ols are the most important compounds in determining the astringency strength of wine [10,57]; in this case, however, flavan-3-ols were more pronounced in Hongsipu wines, albeit no significant difference was observed (Appendix A). Hydroxybenzoic acids were also proved to contribute to astringency [58], and this might be one of the reasons for the stronger astringency of Hongsipu wines. Lower pH and ethanol in Hongsipu wines (Appendix A) could also accentuate the astringent sensation in month [59], and could lead to a higher astringency strength result. In addition, Hongsipu wines showed lower tannin texture than the others, which was basically consistent with the CATA results, possibly due to more flavan-3-ols and hydroxybenzoic acids, as well as lower pH and ethanol level, as suggested [60].

## 4. Conclusions

In this study, primary phenolic compounds, visual properties and mouthfeel of 71 Cabernet Sauvignon wines from the six sub-regions of EFHM were analysed. Through the mining of the OPLS-DA model, it was found that 32 phenolic compounds could be used as characteristic compounds to distinguish wines from different sub-regions. The quantitative analysis results showed that the concentration of phenolic compounds in wines from different sub-regions had their own characteristics; especially, the Hongsipu wines showed great differences in phenolic compounds compared with others. In addition, these characteristics are also reflected in the senses, forming the unique visual properties and mouthfeel of the wines of different sub-regions. 

Thus, a terroir effect was observed for phenolic compounds and detailed studies on the effects of terroir on the phenolic compounds in wines from different sub-regions in EFHM should be further investigated. It should also be noted that the exploration of terroir conditions in different sub-regions of EFHM is still limited, and the typical characteristics of wines in different regions are not well understood. In the future, further probing could be carried out in this respect.

## Figures and Tables

**Figure 1 foods-12-01081-f001:**
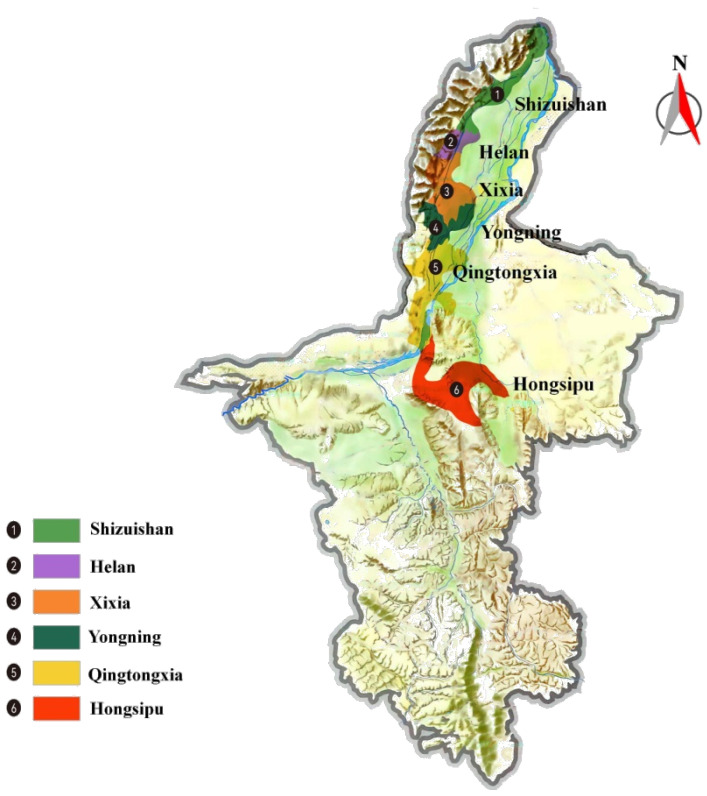
The topographic map of the six sub-regions from EFHM in Ningxia Hui Autonomous Region.

**Figure 2 foods-12-01081-f002:**
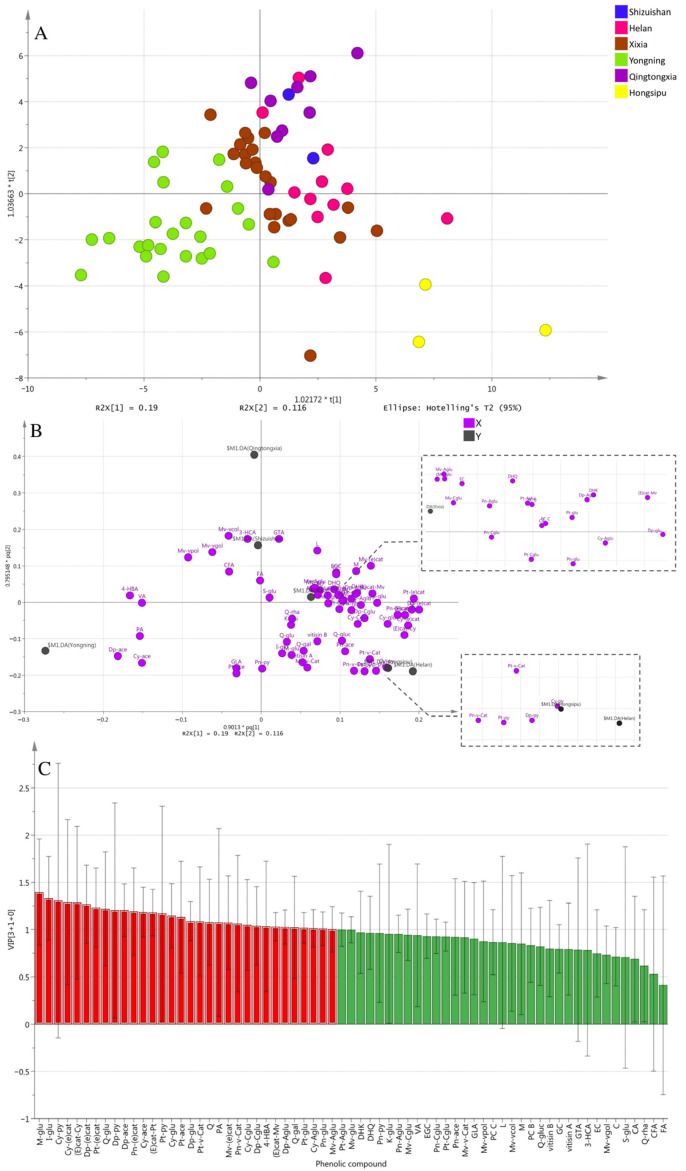
OPLS-DA model based on the concentrations of phenolic compounds in Cabernet Sauvignon wines from six sub-regions of EFHM. (**A**) Score plot (**B**) Loading plot (**C**) VIP plot of the OPLS-DA model.

**Figure 3 foods-12-01081-f003:**
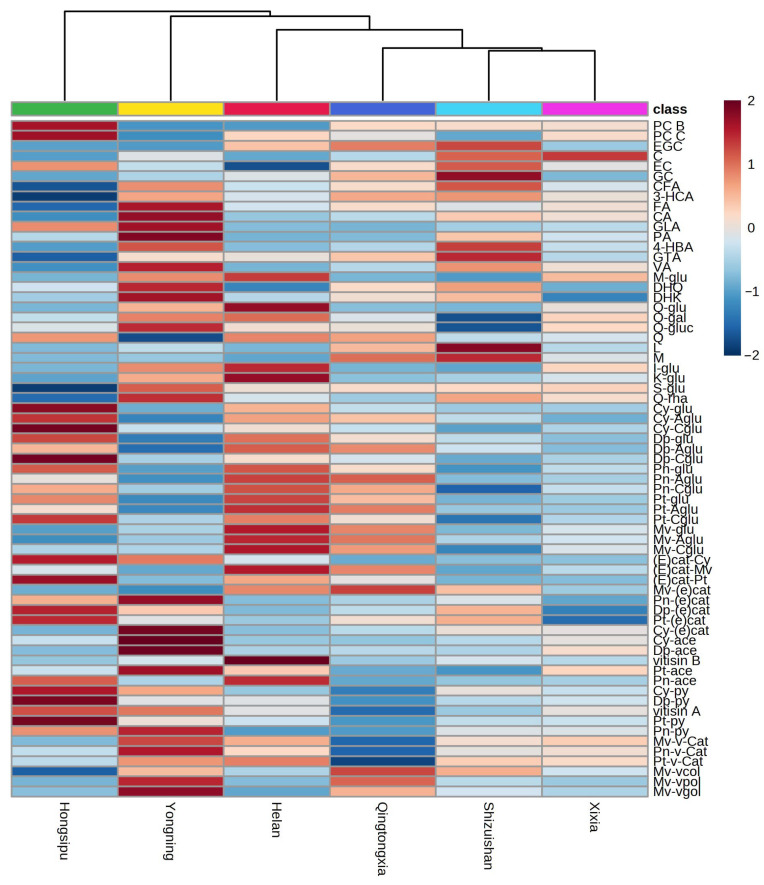
Hierarchical cluster analysis based on the concentrations of phenolic compounds in Cabernet Sauvignon wines in the six sub-regions of EFHM.

**Figure 4 foods-12-01081-f004:**
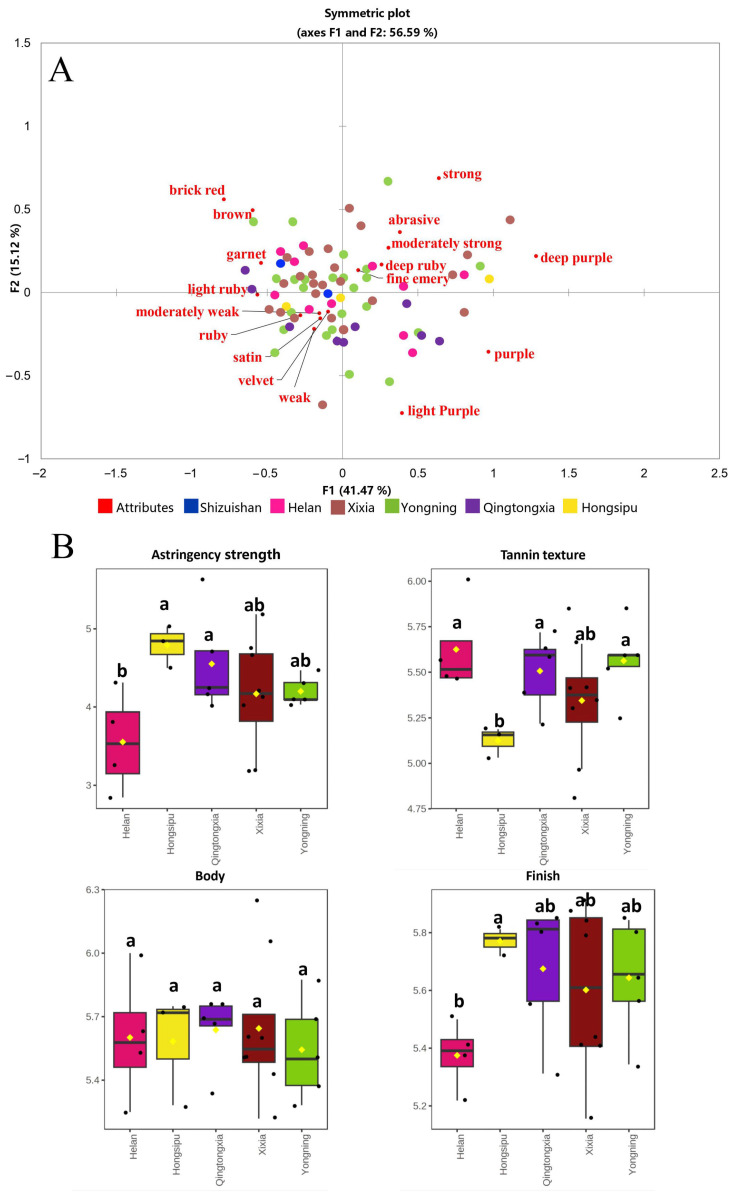
Sensory characteristics of Cabernet Sauvignon wines from the six sub-regions of EFHM. (**A**) Correspondence analysis of wines based on CATA; (**B**) Box plot of wines based on QDA. Different letters in the same area indicate significant difference (*p* < 0.05) using Duncan’s multiple range test.

**Table 1 foods-12-01081-t001:** Abbreviation, MRM information, and calibration curves of phenolic compounds.

Phenolic Compounds	Abbreviation	MRM Transition Ions (m/z)	Retention Time (Min)	Quantitative Standards	Calibration Curves (mg/L)	R^2^
Procyanin B	PC B	577–407	7.6	C	y = 0.0019x − 1.5545	R² = 0.9934
Procyanin C1	PC C	865–407	10.36	C	y = 0.0019x − 1.5545	R² = 0.9934
Epigallocatechin	EGC	305–125	5.2	C	y = 0.0019x − 1.5545	R² = 0.9934
Catechin	C	289–123	5.8	C	y = 0.0019x − 1.5545	R² = 0.9934
Epicatechin	EC	289–123	9.3	EC	y = 0.002x − 2.4286	R² = 0.9929
Gallo-catechin	GC	305–125	2.5	C	y = 0.0019x − 1.5545	R² = 0.9934
Caffeic acid	CFA	179–135	7.2	CFA	y = 0.00008x + 0.8751	R² = 0.9921
3-hydroxycinnamic acid	3-HCA	163–119	10.58	3-HCA	y = 0.000006x + 0.066	R² = 0.9966
Ferulic acid	FA	193–134	12.72	FA	y = 0.0003x + 0.3138	R² = 0.9991
Chlorogenic acid	CA	353–191	6.3	CA	y = 0.00009x + 0.3061	R² = 0.9969
Gallic acid	GLA	169–125	1.7	GLA	y = 0.0002x + 0.4347	R² = 0.9972
Protocatechuic acid	PA	153–109	3.0	PA	y = 0.0002x − 0.0369	R² = 0.9996
4-hydroxybenzoic acid	4-HBA	137–93	5.02	4-HBA	y = 0.0003x − 0.0372	R² = 0.9909
Gentisic acid	GTA	153–109	5.0	GTA	y = 0.0002x + 0.045	R² = 0.9996
Vanillic acid	VA	167–152	6.9	GTA	y = 0.0002x + 0.045	R² = 0.9996
Myricetin-glucoside	M-glu	479–316	13.3	DHQ	y = 0.0002x + 0.2793	R² = 0.9965
Dihydro-quercetin	DHQ	303–125	13.5	DHQ	y = 0.0002x + 0.2793	R² = 0.9965
Dihydro-kampferol	DHK	287–259	17.13	DHQ	y = 0.0002x + 0.2793	R² = 0.9965
Quercetin-glucoside	Q-glu	463–300	16.3	DHQ	y = 0.0002x + 0.2793	R² = 0.9965
Quercetin-galactoside	Q-gal	463–300	15.75	DHQ	y = 0.0002x + 0.2793	R² = 0.9965
Quercetin-glucuronide	Q-gluc	477–301	15.9	DHQ	y = 0.0002x + 0.2793	R² = 0.9965
Quercetin	Q	301–151	23.9	DHQ	y = 0.0002x + 0.2793	R² = 0.9965
Larictrin	L	331–151	24.8	DHQ	y = 0.0002x + 0.2793	R² = 0.9965
Myricetin	M	317–151	19.02	DHQ	y = 0.0002x + 0.2793	R² = 0.9965
Isorhamnetin-glucoside	I-glu	477–314	19.77	DHQ	y = 0.0002x + 0.2793	R² = 0.9965
Kaempferol-3-*O*-glucoside	K-glu	447–255	18.9	DHQ	y = 0.0002x + 0.2793	R² = 0.9965
Syringetin-glucoside	S-glu	507–344	20.1	DHQ	y = 0.0002x + 0.2793	R² = 0.9965
Quercetin-rhamnoside	Q-rha	447–300	19.0	DHQ	y = 0.0002x + 0.2793	R² = 0.9965
Cyanidin-3-*O*-glucoside	Cy-glu	449–287	4.5	Mv-glu	y = 0.00002x + 0.0327	R² = 0.9954
Cyanidin-3-*O*-acetylglucoside	Cy-Aglu	491–287	5.69	Mv-glu	y = 0.00002x + 0.0327	R² = 0.9954
Cyanidin-3-*O*-coumaroylglucoside (cis+trans)	Cy-Cglu	595–287	6.43	Mv-glu	y = 0.00002x + 0.0327	R² = 0.9954
Delphinidin-3-*O*-glucoside	Dp-glu	465–303	4.6	Mv-glu	y = 0.00002x + 0.0327	R² = 0.9954
Delphinidin-3-*O*-acetylglucoside	Dp-Aglu	507–303	5.39	Mv-glu	y = 0.00002x + 0.0327	R² = 0.9954
Delphinidin-3-*O*-coumaroylglucoside (cis+trans)	Dp-Cglu	611–303	6.16	Mv-glu	y = 0.00002x + 0.0327	R² = 0.9954
Peonidin-3-*O*-glucoside	Pn-glu	463–301	5.07	Mv-glu	y = 0.00002x + 0.0327	R² = 0.9954
Peonidin-3-*O*-acetylglucoside	Pn-Aglu	505–301	6.08	Mv-glu	y = 0.00002x + 0.0327	R² = 0.9954
Peonidin-3-*O*-coumaroylglucoside (cis+trans)	Pn-Cglu	609–301	6.76	Mv-glu	y = 0.00002x + 0.0327	R² = 0.9954
Petunidin-3-*O*-glucoside	Pt-glu	479–317	4.7	Mv-glu	y = 0.00002 x + 0.0327	R² = 0.9954
Petunidin-3-*O*-acetylglucoside	Pt-Aglu	521–317	5.76	Mv-glu	y = 0.00002x + 0.0327	R² = 0.9954
Petunidin-3-*O*-coumaroylglucoside (cis + trans)	Pt-Cglu	625–317	6.47	Mv-glu	y = 0.00002x + 0.0327	R² = 0.9954
Malvidin-3-*O*-glucoside	Mv-glu	493–331	5.15	Mv-glu	y = 0.00002x + 0.0327	R² = 0.9954
Malvidin-3-*O*-acetylglucoside	Mv-Aglu	535–331	6.08	Mv-glu	y = 0.00002x + 0.0327	R² = 0.9954
Malvidin-3-*O*-coumaroylglucoside (cis + trans)	Mv-Cglu	639–331	6.74	Mv-glu	y = 0.00002x + 0.0327	R² = 0.9954
Malvidin-3-*O*-glucoside-(epi)catechin (A type)	Mv-(e)cat	783–343	10.53	Mv-glu	y = 0.00002x + 0.0327	R² = 0.9954
Peonidin-3-*O*-glucoside-(epi)catechin (A type)	Pn-(e)cat	753–313	10.29	Mv-glu	y = 0.00002x + 0.0327	R² = 0.9954
Delphinidin-3-*O*-glucoside-(epi)catechin (A type)	Dp-(e)cat	755–315	8.08	Mv-glu	y = 0.00002x + 0.0327	R² = 0.9954
Petunidin-3-*O*-glucoside-(epi)catechin (A type)	Pt-(e)cat	769–329	9.1	Mv-glu	y = 0.00002x + 0.0327	R² = 0.9954
Cyanidin-3-*O*-glucoside-(epi)catechin (A type)	Cy-(e)cat	739–299	8.9	Mv-glu	y = 0.00002x + 0.0327	R² = 0.9954
(Epi)catechin-cyanidin-3-*O*-glucoside (B type)	(E)cat-Cy	737–575	6.39	Mv-glu	y = 0.00002x + 0.0327	R² = 0.9954
(Epi)catechin-malvidin-3-*O*-glucoside (B type)	(E)cat-Mv	781–619	6.97	Mv-glu	y = 0.00002x + 0.0327	R² = 0.9954
(Epi)catechin-petunidin-3-*O*-glucoside (B type)	(E)cat-Pt	767–605	6.5	Mv-glu	y = 0.00002x + 0.0327	R² = 0.9954
Cyanidin-3-*O*-glucoside-acetaldehyde	Cy-ace	473–311	8.5	Mv-glu	y = 0.00002x + 0.0327	R² = 0.9954
Delphinidin-3-*O*-glucoside-acetaldehyde	Dp-ace	489–327	7.2	Mv-glu	y = 0.00002x + 0.0327	R² = 0.9954
Malvidin-3-*O*-glucoside-acetaldehyde	Vitisin B	517–355	10.7	Mv-glu	y = 0.00002x + 0.0327	R² = 0.9954
Petunidin-3-*O*-glucoside-acetaldehyde	Pt-ace	503–341	10.1	Mv-glu	y = 0.00002x + 0.0327	R² = 0.9954
Peonidin-3-*O*-glucoside-acetaldehyde	Pn-ace	487–325	10.14	Mv-glu	y = 0.00002x + 0.0327	R² = 0.9954
Cyanidin-3-*O*-glucoside-pyruvic acid	Cy-py	517–355	8.36	Mv-glu	y = 0.00002x + 0.0327	R² = 0.9954
Delphinidin-3-*O*-glucoside-pyruvic acid	Dp-py	533–371	7.6	Mv-glu	y = 0.00002x + 0.0327	R² = 0.9954
Malvidin-3-*O*-glucoside-pyruvic acid	Vitisin A	561–399	10.415	Mv-glu	y = 0.00002x + 0.0327	R² = 0.9954
Petunidin-3-*O*-glucoside-pyruvic acid	Pt-py	547–385	8.7	Mv-glu	y = 0.00002x + 0.0327	R² = 0.9954
Peonidin-3-*O*-glucoside-pyruvic acid	Pn-py	532–369	9.86	Mv-glu	y = 0.00002x + 0.0327	R² = 0.9954
Malvidin-3-*O*-glucoside-4-vinyl(epi)catechin	Mv-v-Cat	805–643	20.69	Mv-glu	y = 0.00002x + 0.0327	R² = 0.9954
Peonidin-3-*O*-glucoside-4-vinyl(epi)catechin	Pn-v-Cat	775–613	20.51	Mv-glu	y = 0.00002x + 0.0327	R² = 0.9954
Petunidin-3-*O*-glucoside-4-vinyl(epi)catechin	Pt-v-Cat	791–629	19.34	Mv-glu	y = 0.00002x + 0.0327	R² = 0.9954
Malvidin-3-*O*-glucoside-4-vinylcatechol	Mv-vcol	625–463	20.9	Mv-glu	y = 0.00002x + 0.0327	R² = 0.9954
Malvidin-3-*O*-glucoside-4-vinylphenol	Mv-vpol	609–447	21.21	Mv-glu	y = 0.00002x + 0.0327	R² = 0.9954
Malvidin-3-*O*-glucoside-4-vinylguaiacol	Mv-vgol	639–477	21.31	Mv-glu	y = 0.00002x + 0.0327	R² = 0.9954

**Table 2 foods-12-01081-t002:** Color parameters of Cabernet Sauvignon wines from the six sub-regions of EFHM.

	Shizuishan	Helan	Xixia	Yongning	Qingtongxia	Hongsipu
*L**	51.38 ± 2.36 a	44.06 ± 9.66 a	48.02 ± 10.08 a	48.25 ± 8.85 a	50.03 ± 8.95 a	47.46 ± 12.14 a
*a**	40.84 ± 0.34 b	47.24 ± 7.52 a	43.92 ± 7.71 ab	43.09 ± 6.94 ab	41.98 ± 5.82 ab	45.22 ± 7.68 ab
*b**	28.52 ± 5.22 a	24.05 ± 4.91 b	24.36 ± 5.85 b	21.82 ± 5.11 bc	22.16 ± 3.60 bc	19.78 ± 5.15 c

* Expressed as average value plus and minus standard deviation; different letters in the same row indicate significant difference (*p* < 0.05) using Duncan’s multiple range test.

## Data Availability

The data used to support the findings of this study can be made available by the corresponding author upon request.

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
