# Peer review of "Sub-Regional Variation and Characteristics of Cabernet Sauvignon Wines in the Eastern Foothills of the Helan Mountain: A Perspective from Phenolics, Visual Properties and Mouthfeel"

_foods, 2023, doi:10.3390/foods12051081_

Round 1

Reviewer 1 Report

The manuscript entitled "Sub-regional variation and characteristics of Cabernet Sauvignon wines in the eastern foothill of the Helan Mountain: A perspective from phenolics, visual properties, and mouthful palate" presents an attempt to classify Cabernet sauvignon wines based on the specific chemical composition (phenols) and organoleptic characteristics according to their subregional characteristics. 

Aldo in general, the research is well performed when it comes to the analytical methodology (well described and adequate for this purpose), there are several issues that need to be addressed related to the statistical analysis. 

1. Due to the fact that wines are not from the same year (harvest - ranging from 2015 to 2021), the effect of the year must be properly tested and further evaluation and multivariate analysis of the data must be done according to the significance of the year effect. 

2. due to the known reliability issue of OPLS-DA it must be validated properly, (I suggest that you use PCA for this purpose). It is also clear that many of the markers defined by the OPLS-DA do not show any significant differences between regions using ANOVA. 

All of the abbreviations presented in the manuscript must be explained (not only in the supplement file) for easier understanding. 

Figures 2 B is not possible to read (too many compound names that are not visible and must be improved (or replaced with a table). 

Reviewer 2 Report

Organoleptic features are difficult for analysis and presentation and in my opinion graphic form may be improved to increase possibility of understanding what was shown in the graphs 

Reviewer 3 Report

The manuscript submitted for evaluation concerns the analysis of 71 cabernet sauvignon wines from six sub-regions located at the foot of Mount Helan in the Ningxia Hui Autonomous Region. The studies assessed the visual and taste characteristics of the wines as well as phenolic profiles. Visual and sensory assessments were performed by CIELAB and Check-all-that-apply (CATA), respectively. Qualitative and quantitative analysis of phenolic compounds was performed by HPLC-QqQ-MS/MS technique. The results were compiled using appropriate statistical tools.

According to the reviewer, the manuscript is prepared correctly. The importance of the research undertaken is clearly explained. The individual research goals are clearly defined, the methods of achieving them are well selected and described, and the results are presented without reservations.

The disadvantage of this work is the excessive amount of references. In the opinion of the reviewer, this list should be shortened to the most recent items that are necessary to understand the novelty of this article. Another remark concerns Table S7. It is included in the supplementary materials. According to the reviewer, it should be included in the main part of the manuscript, because it allows to understand the results presented in the manuscript. Nevertheless, the presented equations of calibration curves with a very large share of the intercept (b) in some cases are at least puzzling and require verification.

Round 2

Reviewer 1 Report

I do not agree with the answers presented by the authors. 

Related to point 1. 

Point 1: Due to the fact that wines are not from the same year (harvest - ranging from 2015 to 2021), the effect of the year must be properly tested and further evaluation and multivariate analysis of the data must be done according to the significance of the year effect.

Response 1: As a particular product, wine aging is paramount for wine quality. Some premium wines are not released until they undergo extensive aging. Aging techniques were also an indispensable part of vinification and indeed a reflection of terroir. Therefore, we have collected wine with different vintages to represent this character. As suchwines with vintage between 2015-2021 were collected in purpose, and these wines were commerical available (readily to be released and to drink) in each sub-regions of EFHM. We believe that these samples were representative of each sub-region and could present its own unique terroir conditions, which facilitates the summary of the terroir profile of each region and the sensory characteristics of the wine.

Answer: contribution of the aging to the modification of phenolics is significant and it can have a significant effect on the proper definition of terroir expression. There is no information given on when exactly were the collected samples put on the market and for this reason we can not discuss the quality level of the wines based on the harvest year. One of the researches you have given as an example under point 2. is a good example of presenting the effects of both aging process and the origin on proper wine classification https://doi.org/10.1016/j.jfca.2018.05.010

Point 2: Due to the known reliability issue of OPLS-DA it must be validated properly... 

Response 2: We appreciate your valuable advice. But it needs to be explained that we have tried PCA of this experiment, which can not bring good separation effect, which was possiblely due to the fact that an unsupervised model such as PCA was not able to separate the sample. Finally OPLS-DA was chosen. OPLS-DA was extenstively used in wine explortation and discrimination and are ought to be reliable and suitable for this studie. Many researches are available for reference, such as https://doi.org/10.1016/j.jfca.2018.05.010, https://doi.org/10.1016/j.foodcont.2022.109336, references [33] and [21] in the manuscript. In addition, according to your comments, we checked and modified the marker compounds combined with the results of ANOVA. While incorporating the comments of other professors, Table S7(including abbreviated information) was transferred to the manuscript to make the manuscript easier to understand. The difficult reading part in Figure 2 B has been moderately optimized, as shown in the figure below.

Answer: Exactly as presented in both articles you give as the example for POLS-DA utilization it has to be properly validated, and in your research it is not validated at all. Very low correlations that are presented presented in Figure 2B (Loading plot) among original variables and disciriminant function, are reason why these variables are not good as potential markers for wines from specific subregions - oposite to what you are suggesting. 
